# Usage of Machine Learning Techniques to Classify and Predict the Performance of Force Sensing Resistors

**DOI:** 10.3390/s24206592

**Published:** 2024-10-13

**Authors:** Angela Peña, Edwin L. Alvarez, Diana M. Ayala Valderrama, Carlos Palacio, Yosmely Bermudez, Leonel Paredes-Madrid

**Affiliations:** 1Faculty of Mechanic, Electronic and Biomedical Engineering, Universidad Antonio Nariño, Carrera 7 N 21–84, Tunja 150001, Boyacá, Colombia; angelapena@uan.edu.co; 2Doctorado en Ciencia Aplicada, Universidad Antonio Nariño, Carrera 3 Este N 47 A–15, Bogotá DC 110231, Colombia; 3GIMAC (Modeling, Automation and Control Research Group), Mechatronics Engineering Program, Faculty of Sciences and Engineering, Universidad de Boyacá, Carrera 2A Este N 64–169, Tunja 150003, Boyacá, Colombia; edwalvarez@uniboyaca.edu.co; 4Comprehensive Management of Agro-Industrial Productivity and Services GISPA, Santo Tomas University, Tunja, Av. Universitaria, No. 45-202, Tunja 15003, Boyacá, Colombia; dianisayala2010@gmail.com; 5Faculty of Sciences, Universidad Antonio Nariño, Carrera 7 N 21–84, Tunja 150001, Boyacá, Colombia; carlospalacio@uan.edu.co; 6Universidad Simón Bolívar, Caracas, Venezuela; 13-10143@usb.ve

**Keywords:** conductive polymer composite, force sensing resistor, machine learning

## Abstract

Recently, there has been a huge increase in the different ways to manufacture polymer-based sensors. Methods like additive manufacturing, microfluidic preparation, and brush painting are just a few examples of new approaches designed to improve sensor features like self-healing, higher sensitivity, reduced drift over time, and lower hysteresis. That being said, we believe there is still a lot of potential to boost the performance of current sensors by applying modeling, classification, and machine learning techniques. With this approach, final sensor users may benefit from inexpensive computational methods instead of dealing with the already mentioned manufacturing routes. In this study, a total of 96 specimens of two commercial brands of Force Sensing Resistors (FSRs) were characterized under the error metrics of drift and hysteresis; the characterization was performed at multiple input voltages in a tailored test bench. It was found that the output voltage at null force (*V_o_null_*) of a given specimen is inversely correlated with its drift error, and, consequently, it is possible to predict the sensor’s performance by performing inexpensive electrical measurements on the sensor before deploying it to the final application. Hysteresis error was also studied in regard to *V_o_null_* readings; nonetheless, a relationship between *V_o_null_* and hysteresis was not found. However, a classification rule base on *k*-means clustering method was implemented; the clustering allowed us to distinguish in advance between sensors with high and low hysteresis by relying solely on *V_o_null_* readings; the method was successfully implemented on Peratech SP200 sensors, but it could be applied to Interlink FSR402 sensors. With the aim of providing a comprehensive insight of the experimental data, the theoretical foundations of FSRs are also presented and correlated with the introduced modeling/classification techniques.

## 1. Introduction

Conductive Polymer Composites (CPCs) have attracted a lot of attention recently because of their many promising traits like corrosion resistance, easy access to polymers and fillers, and their excellent mechanical and electrical properties [1,2,3]. The high availability of polymers and fillers is a highly desirable characteristic for cost-sensitive applications, such as motion tracking of human limbs [4], speech recognition [5], and sleeping posture monitoring [6].

Several reviews have explored the wide range of applications for CPCs in both industry and everyday life [1,2,3,7]. As a result, there has been a lot of focus on solving manufacturing challenges [2,7] and improving the overall performance of CPCs in real-world applications [8,9,10,11].

This study focuses on a specific type of sensor made from CPCs: Force Sensing Resistors (FSRs), which change their electrical resistance when exposed to compressive force. FSRs belong to a broader category of CPC-based sensors that includes strain, bending, and gas sensors [12,13].

Since this study focuses on FSRs, it is important to note that their main issues come from the viscoelastic nature of polymers, which leads to two types of errors: drift during constant pressure and hysteresis. These problems currently limit the use of FSRs in applications where accuracy and consistency are critical. In our view, based on a review of the literature, there are three main strategies for improving the performance of FSRs. These strategies can also be applied to other types of CPC-based sensors. The methods are described below.

### 1.1. Manufacturing Method

Since the original mix, stir, and dry method [14,15], there has been a boom in different ways to manufacture CPCs. The most promising methods for improving CPC performance or simplifying production include brush painting [8], additive manufacturing [5,16], coating [17,18], spray/screen printing [19,20], microfluidic preparation [21], laser-related methods [22,23,24], and exposure to magnetic/electric field during composite drying process [25].

### 1.2. Type of Polymer and Filler

This is probably the most active roadmap reported in the literature regarding CPCs to date. A wide range of carbonaceous and metallic fillers are usually combined with insulating and/or conducting polymers in multiple ways [23,25,26,27]; this is performed with the goal of obtaining a synergistic response in the nanocomposite that improves its electromechanical properties [27] and/or provide self-healing capabilities [26]. Carbon nanotubes (CNTs), Graphene Nanoplatelets (GNPs), and Carbon Black (CB) have been reported as the preferred conductive fillers [28]; nonetheless, the usage of metallic nanoparticles is also common, and more recently, conductive polymers, such as PEDOT [8,29] and polyaniline, have also been investigated as fillers [26,30]. Regarding the insulating matrix, the list is also vast, raging from polymers such as rubbers [15], polyurethane [17,22,26], polydimethylsiloxane (PDMS) [23], and, more recently, cotton fabrics [18]. At this point, it is worth mentioning the incorporation of Aerogel and Hydrogel Materials (AHM) to the latest research trends on CPCs [26,29]; the high biocompatibility of AHM is a highly desirable characteristic that enables a straightforward integration to wearable sensors and, in general, to any health monitoring device [31].

### 1.3. Modeling and Compensation, Including Artificial Intelligence (AI) Techniques

This is probably the least explored roadmap for enhancing the performance of CPCs. The electrical conduction mechanisms of CPC have been thoroughly studied since the early stages of CPC development [32,33,34,35,36,37,38]. However, these studies have mainly focused on the physical phenomena that enables current conduction rather than proposing modeling techniques that let us compensate for the CPCs’ errors. To the best of the authors’ knowledge, we can only cite a few studies that have actively incorporated modeling techniques to enhance CPC-based sensors; they are briefly described ahead.

Urban, Ludersdorfer, and van der Smagt [39] employed a probabilistic, nonparametric sensor model based on Heteroscedastic Gaussian Processes (HGP) to compensate for the hysteresis error; HGP are alternatives to the well-kwon Preisach model that requires a lot of computational power [40]; therefore, HGP are promising approaches for compensating hysteresis error in CPC-based sensors. On the other hand, Mersch et al. modeled nonlinear phenomena observed during cyclic loading in strain sensors; the model incorporated a complex arrangement of spring and dampers for the longitudinal and transversal contractions [41]. It must be highlighted that the nonlinear phenomena modeled by Mersch et al. has been previously reported by other authors [42,43]. Nguyen and Chauhan incorporated convolutional neural networks to characterize a microfluidic-based pressure sensor with promising results [44]. The authors agree on the opinion stated by Nguyen and Chauhan in that most of the existing studies are focused on new designs and materials rather than working on modeling/compensating techniques for the already existing sensors. Other representative works within the field can be found in the studies from Boland [45] and Wang et al. [46].

Most of the authors’ previous work can be classified within this modeling/compensation category. To cite a few examples, when capacitance and conductance readings are integrated in a combined model, it has been demonstrated that the resulting force estimation is more accurate than a model based on conductance readings alone [47]. More recently, the application of the Six Sigma methodology was embraced as a tool to enhance the part-to-part repeatability of commercial Force Sensing Resistors (FSRs) [48]. It is the authors’ belief that there is plenty of room to improve the performance of currently available FSRs by relying on modeling and compensation techniques.

### 1.4. Machine Learning and CPC-Based Sensors, a Brief Literature Review

The usage of machine learning (ML) techniques is a relatively new approach within the field of CPC-based sensors. A search on the Scopus database with the keywords “conductive polymer composite & machine learning” returned only 27 entries as of October 2024. The studies found can be classified within the following four categories: research articles that incorporated ML for nanocomposite design and manufacturing, combined usage of ML techniques and strain/stress sensors to develop a final application, simulation studies, and review articles. The most relevant studies of each category are discussed.


*Usage of machine learning techniques for nanocomposite design and manufacturing:*


Vaishnavi Thummalapalli et al. [49] introduced a novel method to identify the optimal printing parameters of a Direct Ink Writing (DIW) manufacturing process. The ink in situ assembled in situ by combining a polymer ink with copper fillers. Machine learning techniques were deployed to optimally determine the printing parameters, such as speed, flow pressure, and filler concentration.

Razavi, Sadollah and Al-Shamiri [50] developed a prediction model for polymer composites on the basis of the Taguchi method, Artificial Neural Networks (ANN), and extreme machine learning. The model enabled them to predict the response factor of the assembled nanocomposites resulting from the vigorous mixing of epoxy resin, CNT, and CB.

Niendorf and Raeymaekers [51] employed a supervised machine learning method to predict microfiber alignment and electrical conductivity. The nanocomposites were assembled from sonication of silver-coated glass microfibers in a photopolymer matrix. The machine learning prediction was successful and the results promising.

The study from Hannigan et al. [52] is also worth mentioning within this category.


*Simulation studies:*


Shah et al. [53] implemented a reduced-order model for predicting the effective stiffness of CNT/Polymer composites and their electrical properties. The model comprised ANN and linear regressions. The simulation results were acceptable when predicting the mechanical properties, but the electrical simulations lacked accuracy.

Shi et al. [54] implemented a ML model to optimize nanocomposite materials aimed for electromagnetic interference shielding. The data for this study were taken from already published studies that demonstrate forecasting capabilities. The ML model combined multiple interesting features such as a weighted average ensemble and a 5-fold cross-validation scheme.

The study from Cao and Zhang [55] is also worth mentioning within this category.


*Combined usage of ML techniques and strain/stress sensors to develop a final application:*


The usage of ML techniques is a proven method to merge data from multiple strain/stress sensors; this is the case of the studies from Zhou et al. [56], and Yao et al. [57]. The study from Wang et al. [58] is also worth mentioning within this category.


*Review articles:*


Multiple authors have discussed about the emerging use of ML techniques in the design, manufacturing, and deployment of polymer nanocomposites into a final application; this is the case of the reviews from Wang et al. [59], Sharma et al. [60], Xu et al. [61], and Gao et al. [62].

### 1.5. Aim of This Study and Methodology

The purpose of this study is to use machine learning techniques to classify and/or predict the performance of FSRs by relying upon measurements of sensors’ output voltage at null force (*V_o_null_*). Given the trained machine learning model, it is here demonstrated that *V_o_null_* readings can predict the sensor’s drift and provide some insights on the sensor’s hysteresis. By doing this, final sensor users are capable of choosing in advance the sensors with the better performance and discarding those with high error metrics. It must be highlighted that performing *V_o_null_* readings have the advantage of being simple, non-expensive, and non-reliant on custom mechanical setup.

The rest of this paper is organized as follows: Section 2 discuss about the error metrics commonly found in FSRs, a theoretical model for the conduction and sensing mechanism of FSRs are also presented. The experimental setup is presented in Section 3 together with the test protocol. Section 4 discusses about the experimental results, the machine learning model, and the main findings. A discussion is presented in Section 5. Henceforth, in this manuscript, we no longer use the designation CPC-based sensor, and, instead, we use FSR and sensor indistinctively.

## 2. Theoretical Foundations and Error Metrics of FSRs

This section discusses the theoretical foundations of FSRs and the types of errors originated from polymer viscoelasticity.

### 2.1. Theoretical Foundations of Quantum Tunneling and Constriction Resistance

Quantum Tunneling (QT) and percolation are the predominant conduction mechanisms of FSRs [32,33,34,35,36,37,38]. QT is defined as the ability of electrons to hop from one conducting particle to another despite being separated by a thin insulating layer, i.e., the polymer matrix. An intricate network of neighboring conductive particles creates tunneling paths as shown in Figure 1. On the other hand, percolation occurs when electrons flow through particles in direct contact. This phenomenon is later addressed in this section.

In practice, FSRs are manufactured from a blend of insulating polymer with randomly dispersed conductive nanoparticles. Section 1.1 introduced the most popular preparation routes of FSRs. Similarly, Section 1.2 discussed the most common materials considered as the insulating polymer and the conductive nanoparticles. Regardless of the preparation method and/or materials employed, the sketch of Figure 1 can be taken as the representation of a FSR with a sandwich-like configuration before and after loading.

Due to the externally applied force, conductive nanoparticles come closer, and therefore the sensor current (*I_FSR_*) increases, as predicted by the Simmons’ equations formulated back in the 1960s [63]. The work from Simmons’ models QT conduction using piecewise equations with transitions defined by the externally applied voltage. Therefore, QT conduction is a phenomenon that is highly dependent on the applied voltage, so it is the tunneling resistance (*R_tun_*).

Previous authors’ work has discussed the role of QT and percolation in the sensing dynamics, sensitivity, and part-to-part repeatability of FSRs [48]. A thorough formulation on the underlying physics of FSRs is out of the scope of this article and has been previously addressed by the authors [48]. For space constraints, we only present here the main highlights of such a formulation.

By recalling QT and percolation as the sensing mechanisms of FSRs, the following model can be proposed as the sensor total resistance (*R_FSR_*):*R_FSR_* = *R_tun_* + *R_c_*(1)
where *R_tun_* and *R_c_* are the tunneling and constriction resistance, respectively. The sketch of Figure 1 shows an intricate net of series and parallel connected *R_tun_* and *R_c_* along the nanocomposite. Given an external applied voltage to the nanocomposite (*V*), *R_FSR_* can be measured using the Ohms’ law, i.e., *R_FSR_* = *V*/*I*_FSR_.

Conduction through percolation occurs when particles are in direct contact with each other; see the gray arrow marks in Figure 1. Given the small contact area between particles in contact, the contact resistance is quantized [64]. In this case, a constriction resistance occurs at the interface between them. The theoretical foundations for this phenomenon are beyond the scope of this article, but they can be reviewed in the work of Sattar, Fostner, and Brown [36], and in the study by Shi et al. [65]. When multiple percolation paths are formed along the nanocomposite, it is possible to formulate a power law that relates the applied force with the so-called contact resistance, *R_c_*. Considering that the contact resistance originates from pure mechanical constriction, *R_c_* is a voltage-independent phenomenon.

### 2.2. Theoretical Foundations of Piezoresistivity in FSRs

For a constant applied force, the total sensor resistance, *R_FSR_*, is heavily dependent on the input voltage; this has been reported by multiple authors through current (*I*)–voltage (*V*) plots [14,66]. Changing *V* modifies the proportion of *R_tun_* and *R_c_*; this is because *R_tun_* is a voltage-dependent phenomenon, but *R_c_* is voltage-independent. Therefore, we can tune the applied voltage to make either *R_tun_* or *R_c_* more relevant for a given sensor.

When an external force is applied, *R_FSR_* is modified by a 2-fold mechanism, *R_tun_* is reduced due to diminishing interparticle separation; see Figure 1. Similarly, *Rc* is reduced due to a greater contact area among neighboring particles in the nanocomposite. However, experimental observations over multiple specimens have demonstrated that, for a given sensor, one sensing mechanism often dominates the other [48]; this can be exemplified from the sketches of Figure 2.

Finally, for the unstressed sensors of Figure 2, we can determine whether each specimen shows the dominance of quantum tunneling or percolation by measuring its resistance at null force. If *R_FSR_* is small relative to the average resistance of all samples, then percolation dominates. In this case, the percolation paths create a sort of short circuit between the electrodes; see Figure 2b. Conversely, if *R_FSR_* is large relative to the average resistance of all samples, then quantum tunneling dominates; this is because *R_tun_* is very sensitive to interparticle separation. In fact, as stated by Simmons [63], *R_tun_* is modified in an exponential fashion by interparticle separation. As later demonstrated in this report, sensors operating on the basis of QT or percolation exhibit remarkable performance differences.

### 2.3. Error Metrics Commonly Found in FSRs

As previously mentioned, the most common errors in FSRs are drift under constant loading and hysteresis. When an FSR is constantly loaded for an extended period, the polymer creeps, and *R_FSR_* does as well. Figure 3a shows the typical drift behavior of sensor conductivity (*σ*), where *σ* = 1/*R_FSR_*. The drift error (*d.e.*) can be computed using the following equation after one hour of constant loading [67]:(2)$$d.e.=\frac{\sigma \left(t=3600\right)-\sigma (t=0)}{\sigma (t=0)}\cdot 100\%$$

On the other hand, hysteresis is calculated as the normalized difference in sensor output when loaded to half the nominal capacity during the loading and unloading stages. An ideal sensor should exhibit the same reading regardless of the loading/unloading direction, but the viscoelasticity of polymers creates a memory-like effect that ultimately leads to hysteresis. Hysteresis error (*h.e.*) can be calculated from the following equation [67]:(3)$$h.e.=\frac{\sigma \left(F=F_{nom/2}^{u}\right)-\sigma \left(F=F_{nom/2}^{l}\right)}{\sigma \left(F=F_{nom}\right)}\cdot 100\%$$
where *F^u^_nom/*2*_* and *F^l^_nom/*2*_* match for half of the nominal force applied during the unloading and loading stages, respectively, with *F_nom_* as the nominal sensor force.

The experimental observations from Section 4 demonstrate that both error metrics, hysteresis and drift, depend on the predominant sensing mechanism occurring in the nanocomposite.

## 3. Experimental Setup

The experimental setup can be divided into four subsections. An overview of the performance metrics of FSRs is presented, followed by the mechanical and the electrical setup. Later, the test protocol is presented.

### 3.1. Performance Metrics of Interlink and Peratech FSRs

In order to obtain statistically representative results, the experimental tests in this study were performed on 48 Interlink FSR402 sensors manufactured by Interlink Electronics (Irvine, CA, USA) [68] and on 48 QTC Peratech SP200 sensors manufactured by Peratech Holdco Limited (Sedgefield, UK) [69].

A comparison between both sensors is presented in Table 1. In general, both sensors offer similar performance metrics and sensing ranges. Unfortunately, it is not possible to determine FSRs’ sensitivity a priori given the random dispersion of conductive nanoparticles in the nanocomposite; see Figure 1. Additionally, FSRs’ sensitivity depends on the sourcing voltage, as the tunneling resistance is modified by the applied voltage. Previous authors’ work has discussed about both consideration for FSRs [48].

### 3.2. Mechanical Setup

The mechanical setup is shown in Figure 4. It consists of a linear stepper motor located on top of the test bench. In order to provide mechanical compliance, a spring was added between the motor and the sensors, which were held in place by custom sensor holders. Force loops were closed using an LCHD-5 load cell manufactured by Omega Engineering (Norwalk, CT, USA). Finally, the bunch of sensors was located at the bottom of the test bench using a sandwich-like configuration capable of handling up to sixteen sensors simultaneously; see Figure 4a. With the aim of avoiding undesired sensor displacement during loading/unloading stages, each sensor holder had a notch on the upper side and a round puck on the bottom; see Figure 4e. The sensors holders and the spring were held in place by means of the element shown in Figure 4d,f; the element also ensured sensor alignment during tests.

Given the sandwich-like configuration of Figure 4a, it is naturally expected that the bottom sensors are more heavily loaded than those on top. However, this is not a problem if they are measured and compensated by software. Moreover, it is desirable that sensors are preloaded at any given time, even at null force. By recalling the operating force range from Table 1, FSRs require at least 0.2 N to yield a valid reading; otherwise, the resulting resistance is too noisy to be measured. Nonetheless, this threshold is a theoretical value from the manufacturer. In practice, FSRs should be loaded with about 0.3 N to yield a repeatable null force reading.

If the linear motor is not exerting any force on the bunch of sensors, we can compute the applied null force by summing the individual masses of each element as follows: spring (20 g), element of Figure 4f (20 g), and sensor holder (2 g). Therefore, the topmost element has a null applied force of 0.39 N, and the bottommost sensor has a null applied force of 0.7 N. In both cases, the low range of the operating null force is exceeded, thus ensuring a repeatable null-force reading. However, it is clear—and somewhat concerning—that *V_o_null_* is measured at nearly twice the force for the bottommost sensor when compared with the topmost sensor; this concern is addressed in the electrical setup.

### 3.3. Electrical Setup

Considering that error metrics were defined in terms of sensors’ conductivity, (see Equations (2) and (3)), the use of a driving circuit capable of measuring *σ* was required. Therefore, we discarded voltage dividers as the driving circuit, and, instead, amplifiers in an inverting configuration were used. Similarly, a time-multiplexed circuit was required to gather data from multiple sensors simultaneously.

Figure 5 shows the diagram of a circuit for driving up to 16 FSRs using the multiplexer ADG444. The measured voltage corresponds to the output of the operational amplifier (*V_o_*), which is proportional to sensor conductivity as defined by the inverting amplifier model.
(4)$$Vo=\frac{-R_{f}}{R_{FSR}}\cdot Vi$$

The input voltage (*V_i_*) was provided by a NI9263 board (National Instruments, Austin, TX, USA). A 16-bit NI9205 board model was employed for analog data acquisition. Both the NI9205 and the NI9263 were installed in a CRIO-9035 system running LabVIEW RT (Version 2016). The feedback resistor was set to 510 Ω in all tests.

Finally, it must be emphasized from Equation (4) that variations in *V_o_null_* originating from different preloading are negligible between the bottommost and topmost sensor; this statement holds for the test bench of Figure 4. Given the information provided in FSRs’ datasheets [68,69] and using *V_i_* = 5 V, a preloaded sensor with 0.39 N yields a *V_o_null_* equal to 72 mV, i.e., *R_FSR_* ≅ 35 KΩ (topmost FSR), whereas for the bottommost sensor preloaded with 0.7 N, an *R_FSR_* ≅ 20 KΩ is obtained with *V_o_null_* = 127 mV. The difference between both *V_o_null_* readings is 55 mV, which is a negligible variation as reported later in Section 4. Nonetheless, it must be clarified that preloading do impact *V_o_null_* readings as early mentioned in Section 3.3, but the influence of preloading is beneficial so that it allows repeatable measurements of *V_o_null_* by reducing the electrical noise.

### 3.4. Test Protocol

Before detailing the test protocol of this study, we must first state its purpose as follows: classify and/or predict the performance of FSRs by relying upon measurements of FSRs’ output voltage at null force (*V_o_null_*). Consequently, our classification/prediction criteria must be based only upon *V_o_null_* readings at multiple input voltages; see *V_i_* in Figure 5. The designation of null force (rest state) stands for *F* = 0 N in the test bench shown in Figure 4. Next, the test protocols are described.

Test protocol for assessing the drift error, *d.e.*: Initially, *V_i_* was set to 1 V with sensors at rest state; this allowed us to register *V_o_null_* for each specimen individually. Later, an external force of 10 N was applied to the bunch of sensors for a duration of one hour. The *d.e.* was computed for each sensor using Equations (2) and (4). This process was repeated for *V_i_* within the range [1 V, 7 V] with step increments of 1 V. This allowed us to obtain a 7 × 48 matrix containing the *d.e.* of 48 specimens at seven input voltages for each sensor brand.

Test protocol for assessing the hysteresis error, *h.e.*: Following the similar pattern as for *d.e.*, *V_o_null_* was initially registered for each specimen at *V_i_* = 1 V. Later, external forces of 10 N, 20 N, and 10 N were applied to the bunch of sensors while registering *V_o_*. The process was repeated for *V_i_* within the range [1 V, 7 V] with step increments of 1 V. This allowed us to obtain a 7 × 48 matrix containing the hysteresis error of 48 specimens at seven input voltages for each sensor brand.

## 4. Experimental Results

This section reports the experimental results of both sensor brands for the drift and hysteresis errors. Depending on the nature of the experimental results, a classifier or a predictor is introduced accordingly.

### 4.1. Drift Error

Figure 6 reports the experimental results of *d.e.* at multiple input voltages for both sensor brands. Regardless of the sensor brand, it can be stated that drift decreases as *V_i_* increases. This has already been reported in previous authors’ work [70] and can be attributed to the fact that increasing *V_i_* reduces *R_tun_*, whereas *R_c_* remains unaltered. Considering that (i) polymer viscoelasticity is the main source of drift and (ii) *R_tun_* is originated from the multiple tunneling paths along the nanocomposite, it is possible to reduce the *d.e.* by reducing the ratio *R_tun_*/*R_c_*. Note that this statement makes no assumption about the predominant sensing mechanism in a given sensor.

In addition to the previous statement, if *V_i_* is held constant, a novel observation from Figure 6 can be performed by noticing that, for larger values of *V_o_null_*, a lower *d.e.* is measured. We can corroborate this observation by computing the Pearson Correlation Coefficient (PCC) for each data set in Figure 6; this is shown in Table 2.

From Table 2, note that an inverse correlation can be identified between *d.e.* and *V_o_null_* at any *V_i_*, and for both sensor brands. Therefore, a predictor for the *d.e.* can be obtained by applying a linear regression to the experimental data. Figure 7 reports three different trendlines for each sensor brand. Naturally, the fit parameters are voltage-dependent, but we have chosen *V_i_* = 5 V since it is a commonly used voltage in microcontrollers and microprocessors. Table 3 reports the fit parameters and the coefficient of determination for each model. The whole set of data points is available as Appendix A to this manuscript, so readers can obtain a fit for any desired *V_i_*. Finally, this whole process is summarized in the flowchart in Figure 8.

By recalling the relationship between *V_o_null_* and *R_FSR_* in Equation (4), we notice from Figure 7 that sensors with low *R_FSR_* exhibit a comparatively lower *d.e.* than those with high *R_FSR_*., i.e., low *R_FSR_* implies large *V_o_null_*. Next, we can relate this statement from the theoretical foundation in Section 2.2: sensors operating on the basis of percolation exhibit a comparatively lower *d.e.*, whereas those specimens operating on the basis of quantum tunneling exhibit a larger *d.e.* As previously mentioned, both phenomena occur simultaneously, but often, one sensing mechanism dominates over the other. Similarly, every specimen exhibits a different degree of percolation/quantum tunneling as their sensing mechanism; therefore, we observe a continuously varying trend as predicted by the models in Figure 7.

From Figure 7 and Table 3, we notice that every proposed model exhibits a similar performance as measured from the coefficient of determination (*R*^2^). But models 1 and 3 are preferred since they predict a low boundary for the *d.e.* On the other hand, it is noteworthy that *R*^2^ notably differs from one brand to another, with higher dispersion occurring in the Peratech SP200 sensors. We can only hypothesize the possible reasons for this behavior: a lower *R*^2^ could be originated from poor part-to-part repeatability or uneven distribution of particles along the nanocomposite. Remarkable differences in the particle count from one specimen to another may also play a role in the relative higher dispersion of *d.e.* in the Peratech sensors. But, ultimately, it is hard to figure out what the reason for this phenomenon is.

### 4.2. Hysteresis Error

Figure 9 reports the experimental results of *h.e.* at multiple input voltages for both sensor brands. Unlike previous data from *d.e.*, it is more challenging to draw conclusions and statements from hysteresis data; therefore, the analysis is split for each sensor brand and input voltage.

#### 4.2.1. Hysteresis Error in Interlink Sensors

First, the inverse correlation between *V_i_* and *h.e.* is also holds, i.e., higher *V_i_* yields lower *h.e.* The underlying basis for this phenomenon is the same for the drift error as previously identified, i.e., a larger *V_i_* reduces *R_tun_*, whereas *R_c_* remains unchanged. Considering that the main source of drift is *R_tun_*, a reduction in *h.e.* is naturally expected for incremental *V_i_*; this statement holds for both sensor brands as reported in Figure 9.

Second, a clear trend or pattern cannot be identified for the relationship between the *h.e.* and *V_o_null_* in Interlink sensors. This observation is confirmed by computing the PCC for the data sets of Figure 9a–c, resulting in −0.096, −0.037, and −0.041, respectively. Near-zero values imply no linear correlation between the variables.

Third, in an attempt to overcome this limitation, a more comprehensive model was conceived by including multiple *V_i_* and *V_o_null_* data simultaneously. By carefully analyzing *V_o_null_* data, it was found that some sensors exhibit a parabolic relationship between *V_o_null_* and *V_i_*, whereas others show a linear response. These dissimilar responses are shown in the plot of Figure 10 with an overlying parabolic trend line in both cases.

Next, the generic formula of a parabola relating *V_o_null_* with *V_i_* is as follows:(5)$$V_{o\_null}=f\cdot V_{i}^{2}+g\cdot V_{i}+h$$
where *f*, *g*, and *h* are coefficients determined from a least-squares fit. We can determine how parabolic or linear a sensor is by computing the quotient *g*/*f*. The larger *g*/*f* is, the more linear *V_o_null_* is in regard to changes in *V_i_*; conversely, a small *g*/*f* quotient shows a quadratic response in *V_o_null_*. By recalling the theoretical foundations of tunneling conduction from Simmons [63], a parabolic behavior can be found in tunneling paths when the input voltages exceeds the height of the potential (insulating) barrier. Therefore, the quotient *g*/*f* can be understood as a measure on how quantum tunneling dominates in a given specimen. Figure 10 shows a comparison of two specimens of Interlink sensors with *g*/*f* calculations.

In order to obtain a predictor for the *h*.*e*., a feedforward neural network was implemented with *V_o_null_* and *g*/*f* as inputs. The network embraced 10 neurons in the hidden layer and used *h*.*e*. as the output. Supervised learning was implemented using a backpropagation algorithm; an 80/20 proportion was used for the training/validation datasets. Unfortunately, the predictor showed poor performance when attempting to predict the *h.e.* with an *R*^2^ = 0.24. A second version of the neural network was trained and validated by providing the following four inputs to the model: *g*, *f*, and *h*, with *h.e*. as the network output. In this second version, the results were slightly better with an *R*^2^ = 0.31, but still far from acceptable levels. Despite all our efforts, a predictor/classifier could not be implemented for the hysteresis error in Interlink sensors. It is clear that additional phenomena occur in Interlink sensors that the *V_o_null_* readings are not reflecting; this ultimately impacts the ability of *V_o_null_* to predict the *h.e.* in this sensor brand.

#### 4.2.2. Hysteresis Error in Peratech Sensors

When examining the *h.e.* data for Peratech sensors, it is noteworthy that a predictor model is not a good fit for the data in Figure 9d–f. This is confirmed by computing the PCC, which are −0.14, −0.09, and −0.07, respectively. Although prediction models do not work well here, the data sets in Figure 9d–f are not randomly dispersed. Instead, they show agglomerations that may be properly classified into clusters on the basis of a machine learning classifier. In order to support this statement, we need to focus on a single subplot next.

Focusing on one subplot, Figure 9d, low readings of *h.e.* were experimentally measured for *V_o_null_* < 0.2 and for *V_o_null_* > 0.4. However, for the readings in the range 0.2 < *V_o_null_* < 0.4, the *h.e.* did not exhibit a clear trend. Therefore, fitting those data points to a polynomial function is not appropriate. The same observation can be made for any *V_i_*, as shown in Figure 9e–f, but naturally, the ranges of *V_o_null_* need to be reconsidered in each case.

In order to determine the most appropriate number of clusters, the elbow method was applied to the *h.e.* data at *V_i_* = 5 V. It should be clarified that any *V_i_* could be chosen for the classification, but *V_i_* = 5 V was chosen in order to be consistent with the previous results from drift error. Figure 11 reports the results from the elbow method; it is clear that the inflection point is two clusters (*k* = 2). This is a naturally expected result as one might think that sensors can be classified in two subgroups, i.e., sensors with high *h.e.* and sensors low *h.e.* However, some practical limitations appear when using two clusters.

Vector quantization was applied using the *k*-means clustering method using *k* values ranging from 2 to 4. The partitioning results are shown on Figure 12, Figure 13 and Figure 14, respectively. The pros and cons of using either cluster numbering are described next. When using *k* = 2 in Figure 12, a valid classification between sensors with low and high hysteresis cannot be made; this is especially noticeable for group 0. Likewise, it is worth noticing that when *k* = 2, *V_o_null_* cannot be used as a source for sensor classification. This is so because when 1.5 V < *V_o_null_* < 2.7 V sensors with high and low *h.e.* are observed simultaneously. Finally, centroid location for group 0 lies between two sensor clusters. This observation suggests that an additional group could provide a better clustering.

On the other hand, when *k* = 3 in Figure 13, the centroid location for each group is notably improved, but two problems remain: it is still difficult to provide a valid classification between sensors with low and high hysteresis, and, also, when 1.5 V < *V_o_null_* < 2.7 V, sensors with both high and low *h.e.* are observed simultaneously. These issues can be partially resolved by adding another group.

Finally, when *k* = 4 in Figure 14, it becomes clear that sensors within groups 0, 2, and 3 exhibit a lower *h.e.* than those in group 1. The threshold for the low/high *h.e.* can be drawn at *h.e.* = 18%. Similarly, the centroids with *k* = 4 are located in the middle of data clusters, which reaffirms the number of clusters chosen. However, the present clustering fails to distinguish between sensors with high/low *h.e.* based solely on *V_o_null_* readings. It is clear that groups 1 and 2 fall within the same category; therefore, it is unavoidable to obtain false negatives when classifying sensors on the basis of *V_o_null_* readings.

The following rule can be stated to provide a classification criterion for Peratech SP200 sensors with low/high *h.e.*: if a given specimen exhibits *V_o_null_* < 1.4 V or if *V_o_null_* > 2.9 V, then *h.e.* of the specimen is expected to be lower than 18%. The specimens that fall within this rule represent 48% of the total population (23 out of 48).

On the other hand, if the output voltage at null force falls within the range 1.4 V < *V_o_null_* < 2.9 V, the specimen is expected to have an *h.e.* > 18%. The proportion of false negatives is 12.5% of the total population (six out of forty-eight), and 24% are the sensors with high *h.e.* (six out of twenty-five). The false negatives correspond to group 2 in Figure 14.

For a better understanding of this whole process, we have summarized it in the flowchart shown in Figure 15. Appendix B presents the Python (Version 3.12, 64-bit) code for the elbow method calculation and the *k*-means classification. Comments are included in key code lines for better comprehension.

## 5. Discussion

The usage of output voltage at null force, *V_o_null_*, has demonstrated that it provides valuable information for assessing in advance the performance metrics of Force Sensing Resistors, FSRs. Measuring *V_o_null_* for each individual specimen has the advantage of being simple and inexpensive; this is because a mechanical test bench is not required, and simple electronics are implemented instead. This ultimately enables final sensor users to classify and/or predict the performance metrics of FSRs.

Regarding the drift error, *d.e.*, it was found that for both sensor brands the *d.e.* followed an inverse relationship with *V_o_null_*. Experimental drift data were successfully fitted to inverse polynomial functions, which allowed us to predict the *d.e.* from *V_o_null_* measurements. It must be highlighted that both the experimental data and the models were in concordance with the theoretical foundations of FSRs: from percolation theory, and given Simmons’ equation for quantum tunneling conduction [63], it was forecasted that when percolation dominates as the sensing mechanism, the contact resistance rules over the tunneling resistance. This condition results in a large *V_o_null_*, and, ultimately, a low *d.e.* is measured. Conversely, if tunneling conduction is the predominant sensing mechanism, the tunneling resistance rules over the contact resistance, and, therefore, a small *V_o_null_* is measured, resulting in a high *d.e*.

Regarding the hysteresis error, *h.e.*, the theoretical foundations of FSRs showed a limited modeling of the phenomenon. Despite multiple attempts, a suitable relationship between *h.e.* and *V_o_null_* could not be found; this observation especially holds for the Interlink sensors that exhibited no correlation between *h.e.* and *V_o_null_*. At this point, it must be accepted that the conduction model of FSRs is incomplete and must be updated to include additional variables and/or physical phenomena.

This study developed a classifier for the hysteresis error of Peratech SP200 sensors. The classifier was developed based on the *k*-means clustering method and allows us to select those sensors that exhibit *h.e.* below and above 18%.

### 5.1. Limitations of the Proposed Method

Previous authors’ work has demonstrated that the proposed model is capable of predicting—up to an extent—the sensitivity of FSRs based upon *V_o_null_* readings [48], and, more recently, this study has also demonstrated that the *d.e.* can be correlated with *V_o_null_* measurements. But when dealing with the *h.e.*, it is clear that additional phenomena are occurring under the surface that are not being taken into account. We can only hypothesize the possible reasons for this mismatch, with uneven particle count being a possible explanation. The conduction model of FSRs was developed under the assumption that the particle count is relatively constant among sensors, but if a given specimen exhibits a considerably lower number of conductive particles, it is foreseeable that viscoelasticity is more predominant in the specimen, thus yielding a larger *h.e.* The opposite is also expected, a larger particle count may reduce viscoelasticity in the nanocomposite, thus reducing the *h.e.* Unfortunately, to the best of the authors knowledge, there is not a known method to estimate the particle count based on electrical measurements.

Following the discussion on particle count, an undesirable phenomenon occurring in FSRs is the excessive agglomeration of conductive nanoparticles in the nanocomposite. This phenomenon is usually a major concern in polymer composites. For this reason, vigorous mixing of the polymer and nanoparticles is required. If excessive agglomeration occurs during sensor assembly, then current conduction becomes inhomogeneous, ultimately affecting overall performance; also, the sensitivity changes dramatically upon the activation/deactivation of the conductive path with excessive agglomeration.

Finally, the authors have identified two possible limitations for the proposed methodology. Other possible limitations may arise in the future, as mentioned in Section 5.3. The currently identified limitations are listed:A possible limitation of the proposed method is the poor part-to-part repeatability of sensors; refer to Table 1 for the repeatability metrics of each sensor model. To use the proposed method, FSRs must be assembled on a highly repeatable assembly line with high-quality standards for the materials employed. If this requirement is not met, a shift in the PCC metrics reported for the *d.e*. may occur. A similar degradation is expected for the *h.e.* classifier if part-to-part repeatability is low.A logical concern resulting from the proposed method is its applicability to other sensor brands and, ultimately, to different types of sensors made from polymer composites. This concern is indeed a focus for the authors’ future work, as described later in Section 5.3. However, we can hypothesize on the minimum requirements for applying the proposed methodology. First, it is required to consider whether the sensor operates based on quantum tunneling and percolation, keeping in mind that these criteria were our starting point in Section 2. Some polymer-based sensors operate on different principles, such as Fabry–Pérot, which may not be suitable for our proposed method. Secondly, we must address whether the sensor provides an output voltage/resistance/capacitance when unloaded. Keep in mind that *V_o_null_* is the only classification criterion for the proposed method. If these two questions are positively replied, then the odds of the method functioning are high.

### 5.2. Comparing the Proposed Methodology with Existing Literature

According to the existing literature on ML and polymer composites, the proposed method targets similar objectives as those in previously published studies, such as Vaishnavi Thummalapalli et al. [49], Razavi, Sadollah and Al-Shamiri [50], Niendorf and Raeymaekers [51], and Hannigan et al. [52]. These studies focused on predicting and modeling the mechanical and electrical properties of nanocomposites. Our study instead targeted performance metrics, which are somewhat related to the existing literature, but it maintains originality. This observation also applies to the simulation studies by Shah et al. [53], Shi et al. [54], and Cao and Zhang [55], with the difference that these studies are supported by third-party experimental results. Nonetheless, given the vast literature on polymer composites, it is a reasonable approach to conduct research on already published results.

Finally, it should be emphasized that both the *d.e.* and *h.e.* exhibit an inverse correlation with the input voltage. This observation has been previously reported by the authors and aligns with the foundations of FSRs mentioned earlier [70].

### 5.3. Future Authors’ Work

Building on the authors’ most recent study [48], the application of this methodology to other sensor brands and to commercially available strain sensors is expected. This approach has two main objectives: (1) to validate and/or refute the proposed method and (2) to assist final sensor users to assess FSRs’ sensitivity and performance metrics in advance without the need for a specialized test bench.

## Figures and Tables

**Figure 1 sensors-24-06592-f001:**
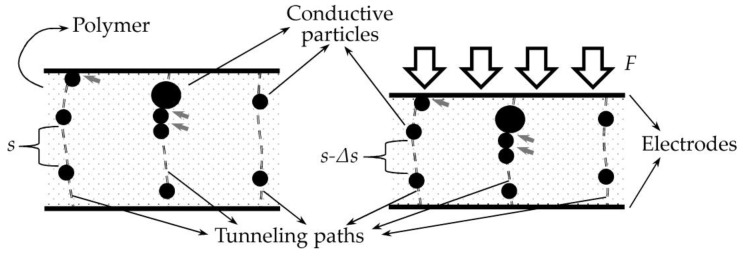
Sketch of an unloaded (**left**) and a loaded (**right**) Force Sensing Resistor. An external applied force (*F*) causes a reduction in the interparticle separation from *s* down to *s*-Δ*s*; this applies to all the tunneling paths along the nanocomposite. The constriction resistance spots are signaled as gray arrow marks. The constriction resistance is also modified by the applied force.

**Figure 2 sensors-24-06592-f002:**
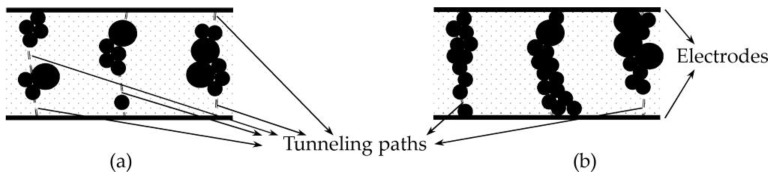
Sketches of FSRs with two different sensing mechanisms. (**a**) Quantum tunneling dominates as agglomerated particles are separated from each other, but connected through multiple tunneling paths. (**b**) Percolation dominates as particles form connection bridged between electrodes; there are only a few tunneling paths along the nanocomposite.

**Figure 3 sensors-24-06592-f003:**
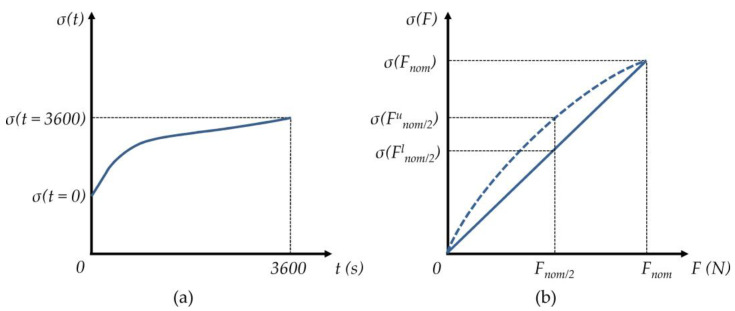
Sketch of common errors in FSRs measured from sensor conductivity (*σ*). (**a**) Drift error occurring after one hour of constant loading. (**b**) Hysteresis error occurring during loading (solid line) and unloading (dashed line) stages.

**Figure 4 sensors-24-06592-f004:**
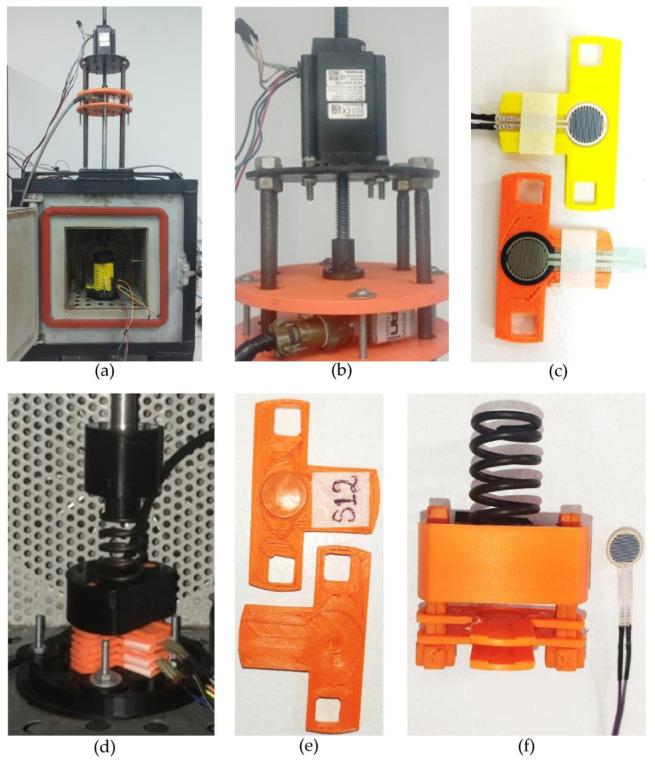
Photographs of the mechanical setup. (**a**) Overview of the testbench; (**b**) Zoom-in photo depicting the linear motor for applying forces to the bunch of sensors. (**c**) Photograph of FSRs installed inside sensor holders, Peratech SP200 (yellow) and Interlink FSR402 (orange). (**d**) Zoom-in photo depicting the sensors inside the chamber and the spring for mechanical compliance. (**e**) Photograph of two side-by-side sensor holders showing the puck (top side) and the notch (bottom side). (**f**) Custom design element for holding aligned the sensor holders, and the spring. A FSR was placed near the element for comparison purposes.

**Figure 5 sensors-24-06592-f005:**
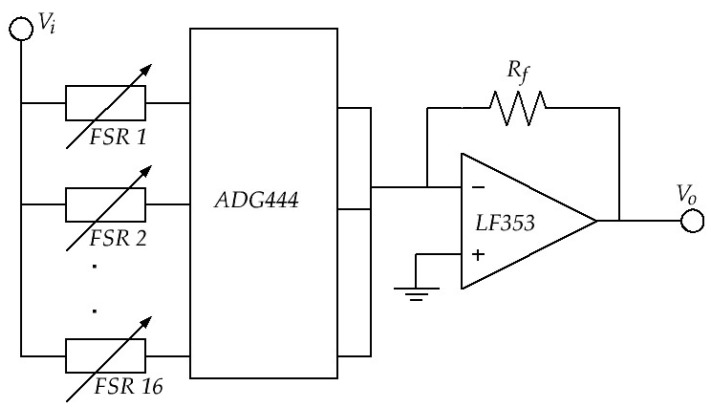
Inverting amplifier for measuring the conductivity of 16 FSRs. The ADG444 was added in order to handle the sensors in a time-multiplexed fashion.

**Figure 6 sensors-24-06592-f006:**
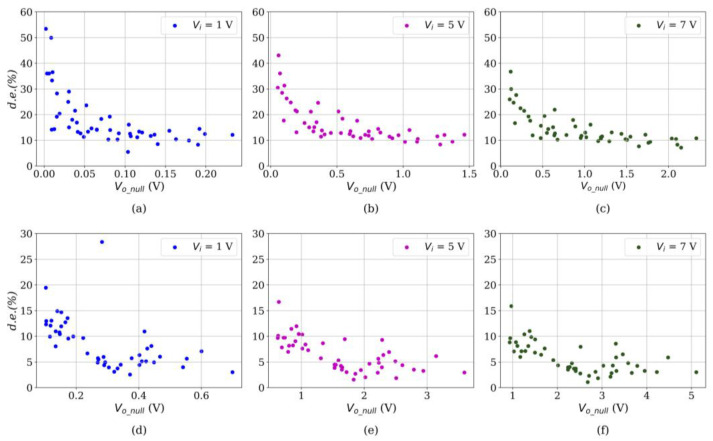
Drift error (*d.e.*) at multiple input voltages for the Interlink FSR402 (**a**–**c**) and QTC Peratech SP200 (**d**–**f**) sensors.

**Figure 7 sensors-24-06592-f007:**
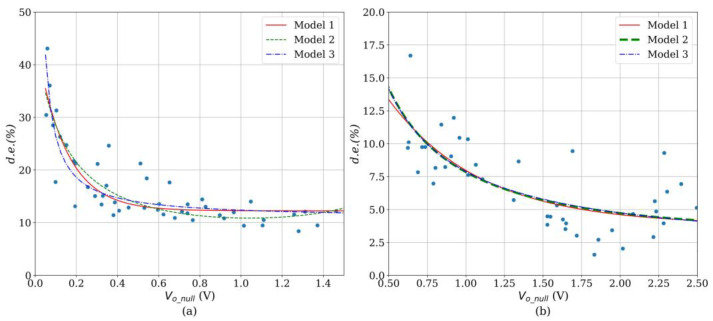
Drift error (*d.e.*) measured at *V_i_* = 5 V with three superimposed trendlines. (**a**) Interlink FSR402, (**b**) QTC Peratech SP200.

**Figure 8 sensors-24-06592-f008:**
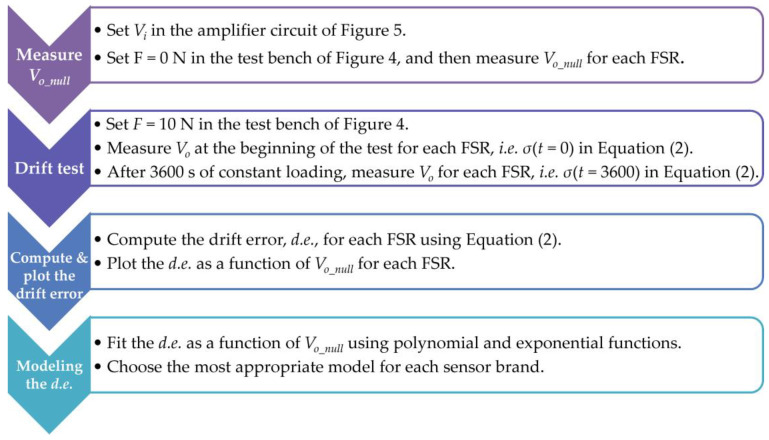
Flowchart summarizing the process for measuring and modeling the drift error in FSRs.

**Figure 9 sensors-24-06592-f009:**
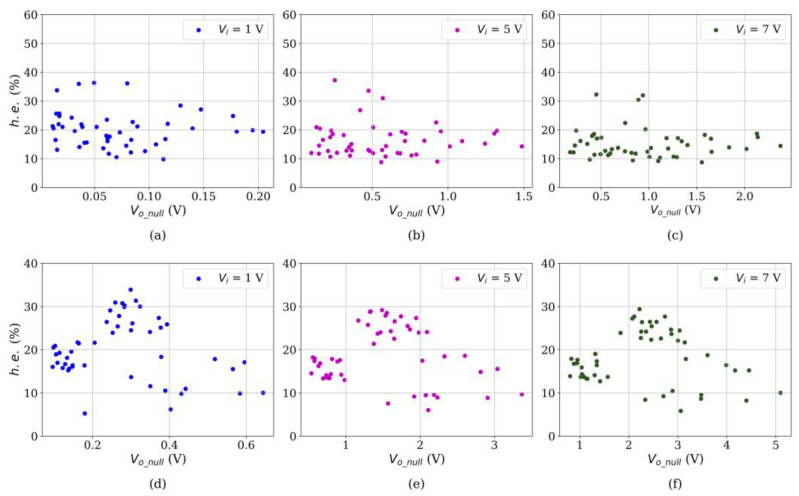
Hysteresis error (*h.e.*) measured at multiple input voltage for Interlink FSR402 (**a**–**c**) and QTC Peratech SP200 (**d**–**f**) sensors.

**Figure 10 sensors-24-06592-f010:**
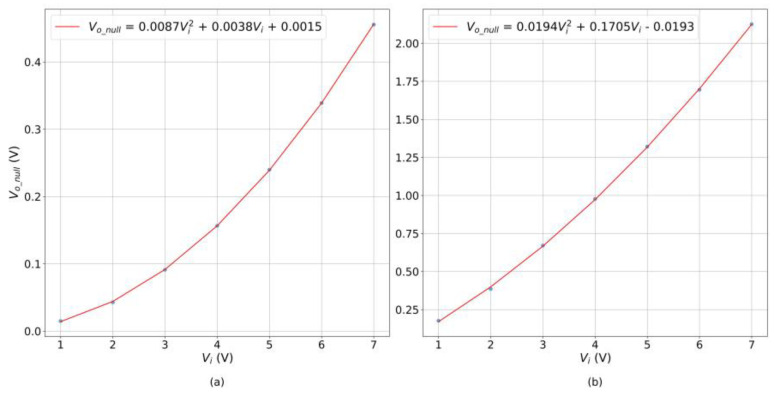
Plots of two Interlink sensors showing the relationship between *V_o_null_* and *V_i_* at null applied force. (**a**) Parabolic behavior with *g*/*f* =0.44. (**b**) Linear behavior with *g*/*f* = 8.78.

**Figure 11 sensors-24-06592-f011:**
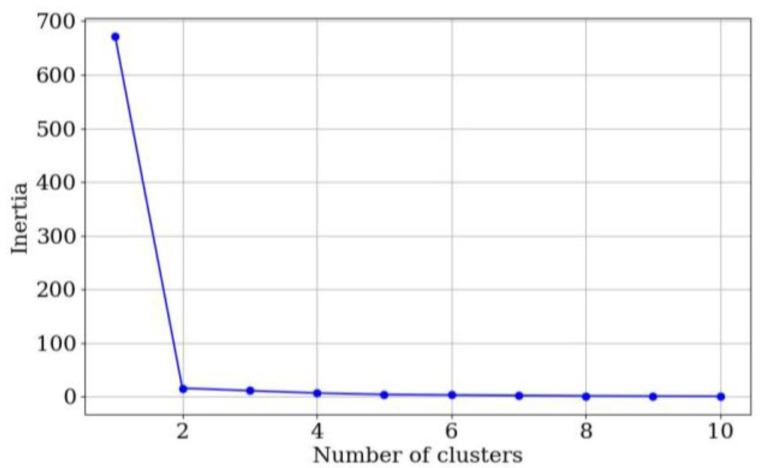
Elbow method results for the *h.e.* data of Peratech sensors at *V_i_* = 5 V.

**Figure 12 sensors-24-06592-f012:**
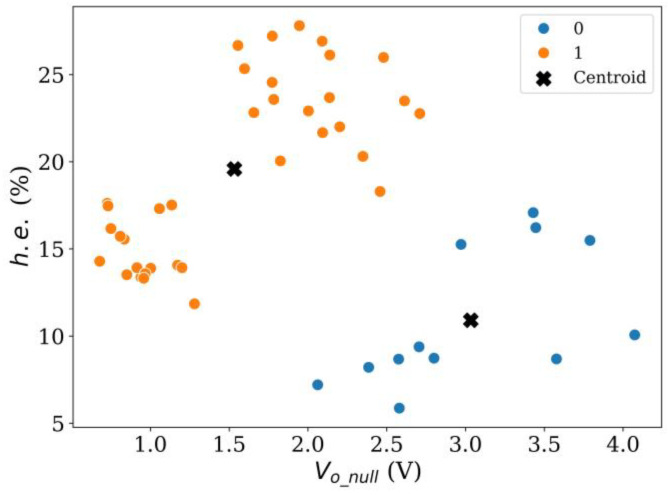
Result of *k*-means clustering method using *k* = 2 for Peratech SP200 sensors at *V_i_* = 5 V.

**Figure 13 sensors-24-06592-f013:**
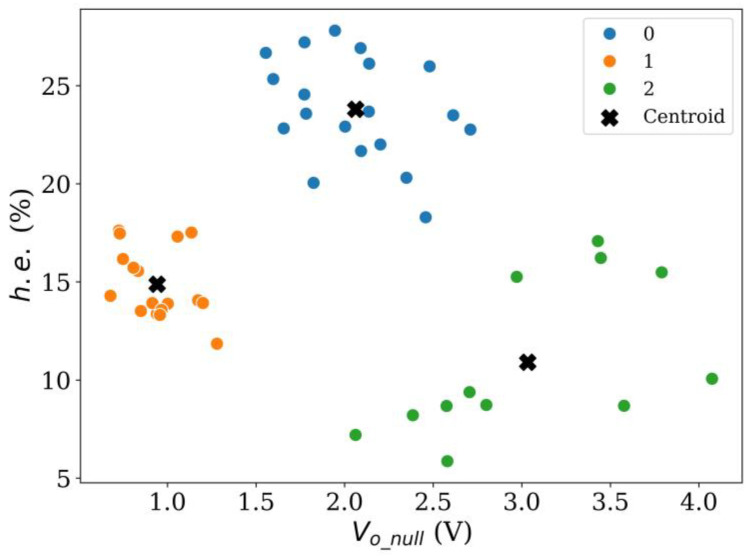
Result of *k*-means clustering method using *k* = 3 for Peratech SP200 sensors at *V_i_* = 5 V.

**Figure 14 sensors-24-06592-f014:**
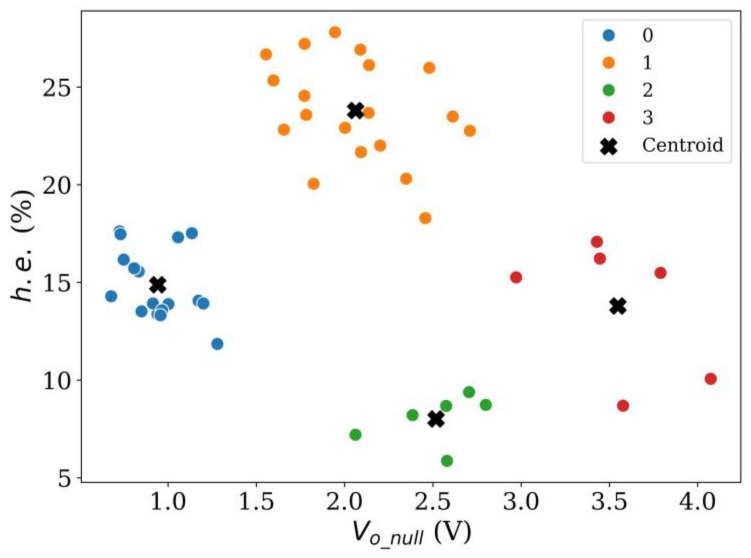
Result of *k*-means clustering method using *k* = 4 for Peratech SP200 sensors at *V_i_* = 5 V.

**Figure 15 sensors-24-06592-f015:**
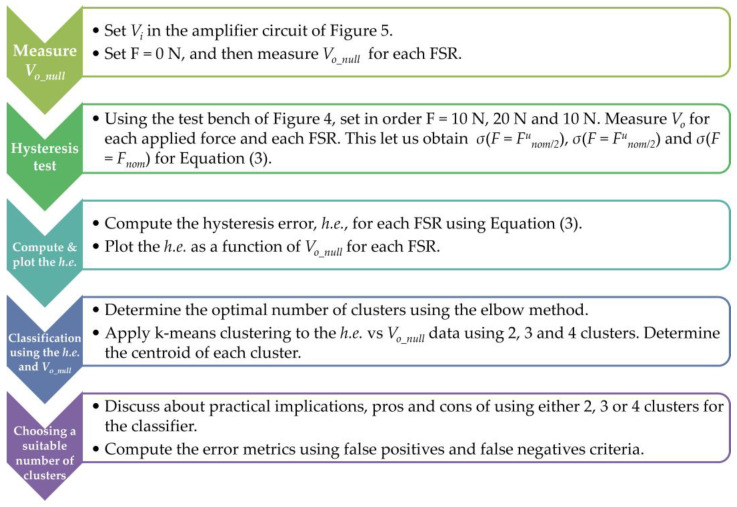
Flowchart summarizing the process for measuring and assessing the hysteresis error in Peratech SP200 sensors.

**Table 1 sensors-24-06592-t001:** Metric comparison between Interlink and Peratech sensors.

Error Metric/Parameter	Interlink FSR402 [68]	QTC Peratech SP200 [69]
Operating force range	0.2 N to 20 N	0.1 N to 20 N
Part to part repeatability	±6%	<4.5%
Mechanical sensing diameter (Active Sensing Area)	1.82 cm (2.6 cm^2^)	1 cm (0.78 cm^2^)
Hysteresis error (%)	10%	8.5%
Drift error (%)	<5% per log (time)	<2% per log (time)

**Table 2 sensors-24-06592-t002:** Pearson Correlation Coefficient (PCC) for the drift error as a function of *V_o_null_* at multiple input voltages.

Sensor Brand	*V_i_* = 1 V	*V_i_* = 5 V	*V_i_* = 7 V
Interlink FSR 402	−0.61	−0.69	−0.72
Peratech SP200	−0.55	−0.67	−0.64

**Table 3 sensors-24-06592-t003:** Fit parameters for the drift error (*d.e.*) models at *V_i_* = 5 V.

Sensor	Parameter	Model 1 $c+a{\cdot e}^{-b\cdot V_{o\_null}}$	Model 2 $\frac{a}{b\cdot {\left(V_{o\_null}\right)}^{2}+c\cdot V_{o\_null}+d}$	Model 3 $a+\frac{b}{V_{o\_null}}$
Interlink FSR 402	a	32.96	182	10.82
	b	7.01	−11.7	1.55
	c	12.24	24.43	--
	d	--	4.06	--
	*R* ^2^	0.7608	0.7567	0.7684
Peratech SP200	a	21.85	0.404	1.59
	b	1.65	−0.0081	6.28
	c	3.81	0.0583	--
	d	--	0.0012	--
	*R* ^2^	0.5843	0.5759	0.5731

## Data Availability

Experimental data are available as an attachment to this manuscript.

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
