# Peer review of "Usage of Machine Learning Techniques to Classify and Predict the Performance of Force Sensing Resistors"

_sensors, 2024, doi:10.3390/s24206592_

Round 1

Reviewer 1 Report

Comments and Suggestions for Authors

This paper presents an alternative and novel method for examining force sensing resistors (FSRs). It uses machine learning techniques to analyze FSRs and forecast their performance based on electrical measurements. The authors analyze the correlation between the output voltage at zero force and the drift error and hysteresis metrics. This research is important because of the growing interest in polymer-based sensors and the potential for machine learning or artificial intelligence techniques to optimize sensor performance. Therefore, the research is relevant and could attract the attention of the scientific and technological community. After making the suggested improvements, I recommend that this paper be accepted for publication in this journal.

The manuscript is well-structured and has a clear introduction, methodology, results, and discussion sections. To enhance the research paper, the following suggestions can be taken into account: 

·         The introduction emphasizes the importance of the research on conductive polymer composites (CPCs) and their applications, making the authors feel valued and significant. However, the manuscript could benefit from a more detailed literature review to provide a comprehensive background on previous work on FSRs and machine learning applications in sensor technology.

·         Include more quantitative data on the sensor's sensitivity, response time, and performance metrics to provide a more detailed analysis of the sensor's capabilities and effectiveness.

·         The authors employed a tailored test bench to characterize 96 specimens from two commercial brands of FSRs. The use of k-means clustering to classify sensors based on Vo_null readings is a novel approach. However, the methodology section lacks sufficient detail regarding the experimental setup, including the specific parameters used for testing and the statistical methods applied for data analysis. A more thorough explanation of the clustering process and the criteria for classifying sensors as high or low hysteresis would enhance the reproducibility of the study.

·         The results indicate a significant inverse correlation between Vo_null and drift error, which is a valuable finding. However, the lack of a similar relationship for hysteresis raises questions about the underlying mechanisms affecting these metrics. This indicates that the initial voltage reading does not provide insights into the hysteresis behavior of the sensors, which is the difference in output when the input is increased versus when it is decreased. The authors should provide a more in-depth discussion of the potential reasons for this discrepancy and how it may impact the practical application of their findings. To substantiate the claims of correlation, the authors should ideally provide statistical metrics such as correlation coefficients (e.g., Pearson or Spearman) and p-values to demonstrate the strength and significance of the relationships observed. The correlation findings are crucial for practical applications. If Vo_null can reliably predict drift, it allows users to select sensors with better performance before deployment. However, the lack of correlation with hysteresis suggests that additional factors may influence hysteresis behavior, warranting further investigation.

·         The classification rule based on k-means clustering is an interesting contribution, but the authors should include more quantitative results to support their claims. For instance, providing metrics such as accuracy, precision, and recall for the classification would strengthen the validity of their approach. Additionally, a comparison with other machine learning techniques could provide insights into the effectiveness of the proposed method.

·         The manuscript briefly touches on the theoretical foundations of FSRs, but this section could be expanded. A more comprehensive discussion of the sensing mechanisms and the materials used in FSRs would provide readers with a better understanding of the context and implications of the findings.

·         The conclusion summarizes the key findings but could be more impactful by discussing the broader implications of the research. The authors should consider addressing potential future work, including the application of their findings in real-world scenarios and the exploration of other machine-learning techniques. The findings regarding correlation can guide future research to explore other variables or machine learning techniques that might better predict hysteresis behavior in FSRs.

·         The references cited in the manuscript are relevant, but the authors should ensure that they include the most recent studies in the field. A more extensive review of the literature would not only strengthen the introduction but also provide a solid foundation for the discussion of results.

The manuscript presents a promising approach to improving the performance of FSRs through machine learning techniques. While the study has significant potential, addressing the aforementioned points will enhance the clarity, rigor, and impact of the research. I recommend the authors revise the manuscript accordingly before resubmission.

Note: In this evaluation report, the reviewer used Grammarly AI to aid in the syntax and compression of the writing. This tool was used directly in Microsoft Word.

Comments on the Quality of English Language

NA

Reviewer 2 Report

Comments and Suggestions for Authors

Overall, this work presents an interesting approach to an important topic. I would like to extend my congratulations to the authors for their efforts. 

My main reservation is that I cannot see how polymer sensors stacked on plastic 3D printed parts, assuming PLA or ABS based, having the force applied axially on the full stack do not get impacted. Is there a cross-reference in the experimentation between a single sensor/ single sensor in plastic holder/single sensor in metal holder for example that justifies that this is not an issue? 

From my understanding based on Fig.4d, the holders are shaped as the sensors, but how certain are the authors that they stay flush? Does the 3D printer used height variance not cause some to protrude and some to rest lower than the plastic casing?

It is crucial that the authors elaborate on that, and in general it would be beneficial to further detail the whole study protocol (in all mechanical, electrical, and data processing parts) put it first, and then analyze how this is implemented in reality, it has a more natural flow. 

However, the following minor points also require some attention:

PDMS is also an abbreviation, like the others which are expanded in the same paragraph

The bibliography/references section needs attention, as by my understanding for some reason it is in Spanish and not English (y instead of and). Also, generally, the et al. in the main text should be used for papers with more than three authors, and up to three they should all be mentioned. 

I think a reference for equations (2) and (3) would be beneficial

There are some quality issues with the graphs. They are of low quality and often pixelated, I am sure they can be improved. 

Also, the overall typography can be improved with some tables and graphs misaligning, or cropping. 

Comments on the Quality of English Language

There are various points in which grammar and editing should be checked, for example:

"The electrical conduction mechanisms of CPC has been [...]",

 "these studies have mainly focused on the physical phenomena that enables current [...]" etc.

Reviewer 3 Report

Comments and Suggestions for Authors

In the manuscript the authors proposed a method to predict the force sensing resistors' performance based on machine learning techniques. The developed strategy is useful and important in improving the practical applications of flexible mechanical sensing devices, thus making a great impact in various related areas, including health monitoring, human-machine interfaces, object/surface recognition, etc. The reviewer considers that this manuscript could be published after the following revisions.

1. The applicable range of the proposed method should be carefully discussed again. Whether all force sensing resistors based on conductive polymer composites could use the proposed strategy? Or whether it is also applicable to other kinds of resistive force sensors?

2. The writing and presentation of the manuscript need to be improved. For example, the introduction should be structured again to enhance the logic. It would be helpful to state the specific problem being addressed more clearly in this section, perhaps by highlighting the challenges with current sensor technologies that this research addresses. As reviewed in the literature (Machine learning-enhanced flexible mechanical sensing. Nano-Micro Letters 15.1 (2023): 55.), various machine learning techniques have been widely applied in the flexible mechanical sensing areas. The authors should refer to them for the latest progress.

3. Adding a diagram or flowchart of the machine learning process could enhance understanding, showing how the data flows from collection through to analysis.

4. When discussing the results of the machine learning analysis, it could be useful to compare your results with existing methods or benchmarks within the literature to highlight the performance improvements or the advantages your approach offers.

Comments on the Quality of English Language

Ensure consistency in the terminology used throughout the manuscript. For example, if you start with "Force Sensing Resistors (FSRs)" make sure to use this term consistently rather than switching to "sensors" or other synonyms unless they are defined. Besides. the language could be improved for easier understanding.

Round 2

Reviewer 1 Report

Comments and Suggestions for Authors

Dear Authors,

I express my sincere gratitude for thoroughly addressing all the suggestions in my initial review. Your modifications have significantly enhanced the quality of the manuscript, improving its clarity and depth.

Your detailed response to each suggestion, particularly the expanded literature review, the inclusion of additional performance metrics, and the elaboration on the experimental setup, has strengthened the overall contribution of this paper. The added sections on machine learning methodology, correlation analysis, and a more comprehensive discussion of the limitations and future work have also significantly improved the manuscript.

I am pleased with the overall result, and I believe the paper is now ready for publication. Your efforts to improve the manuscript have clearly paid off, and it will undoubtedly make a valuable contribution to the field.

Thank you once again for your commitment to refining this work.